

# The syndemic condition of psychosocial problems related to depression among sexually transmitted infections patients

Chen Xu[1,2,*], Yuan Shi[3,*], Xiaoyue Yu[2], Ruijie Chang[2], Huwen Wang[2], Hui Chen[2], Rongxi Wang[2], Yujie Liu[2], Shangbin Liu[2], Yong Cai[2], Yang Ni[1] and Suping Wang[2]

[1] Shanghai Skin Disease Hospital, Shanghai, China
[2] School of Public Health, Shanghai Jiao Tong University School of Medicine, Shanghai, China
[3] Shanghai Jiao Tong University School of Medicine, Shanghai, China
* These authors contributed equally to this work.

## ABSTRACT

**Background**. The prevalence of depression in sexually transmitted infections (STIs) patients is much higher than general public. However, studies focusing on comprehensive psychosocial effects on depression among STIs patients are limited. This study aimed to examine association of multiple psychosocial syndemic conditions with depression among STIs patients in Shanghai, China.

**Methods**. We conducted a cross-sectional study and recruited 910 STIs patients from Shanghai Skin Disease Hospital. Participants self-reported their demographics and themselves completed the scales of depression, self-esteem, loneliness, social support, entrapment, defeat and interpersonal needs. Logistic regressions were performed to detect the possible contributing psychosocial factors for depression and to verify the syndemic conditions of psychosocial problems.

**Results**. Of the STIs patient sample, the prevalence of depression was 17.9%. Multivariable analysis showed low-level self-esteem (odds ratio [ORm]: 2.18, 95% CI [1.19–4.00]) and social support (ORm: 2.18, 95% CI [1.37–3.46]), high-level entrapment (ORm: 6.31, 95% CI [3.75–10.62]) and defeat (ORm: 2.60, 95% CI [1.51–4.48]) increased the risk of depression. Psychosocial syndemic conditions magnified effect in fusing depression (adjusted odds ratio [AOR]: 11.94, 95% CI [7.70–18.53]). Participants with more than 4 psychosocial problems were about 22 times more likely to have depression (AOR: 22.12, 95% CI [13.19–37.09]).

**Conclusions**. The psychosocial problems syndemic magnifying the risk of depression was confirmed and psychosocial interventions to prevent depression is needed among STIs patients.

Corresponding authors
Yang Ni, niyanghj@163.com
Suping Wang,
wangsuping@shsmu.edu.cn

# INTRODUCTION

Sexually transmitted infections (STIs) are spread predominantly by sexual contact and body fluid exchange, among which trichomonas, chlamydia, syphilis, gonorrhea, are most common globally. More than 1 million STIs are acquired every day worldwide

(*Rowley et al., 2019*). In China, acquired immunodeficiency syndrome (AIDS), syphilis, gonorrhea are three major types of the greatest prevalence in China. And both AIDS and syphilis showed upward trends from 2004 to 2016, with an incidence of 3.97/10,000 and 31.97/10,000 in 2016, respectively (*Ye, Liu & Yi, 2019*). STIs patients are at high risk of HIV infections and transmissions due to unprotected sexual encounters. Meanwhile, depressive symptom rates are extremely high among STIs patients. An American study reported 39.2% of STIs patients in the survey had probable depression (*Erbelding et al., 2001*). In 2015, *Mo et al. (2015)*. reported 43% probable cases of depression in Chinese female STIs patients in Hongkong, while the depressed patients accounted for nearly 4.4% and 4.2% of the world and Chinese whole population, respectively. The issue is increasingly pressing.

For STIs patients, depression and status of infection interact as both cause and effect. Researches had showed that patients with sexually transmitted infections bore heavier psychosocial burden than general population, primarily caused by poor self-image and sex-related concerns (*Qi et al., 2014*). Emotional impact was seen most among them. On the other hand, symptoms of depression also concerned with unprotected sexual intercourse, multiple sex partners and gave rise to a potential prevalence of STIs (*Hutton et al., 2004*; *Shrier, Harris & Beardslee, 2002*). Culture difference did not seem to break the correlation. Based on Taiwan National Health Insurance Research Database, *Huang et al. (2018)* conducted a retrospective study in 5,959 depression patients, among which 5.0% was later diagnosed with STIs, compared to 3.2% in control group without depression ($P < 0.001$).

Depression is caused by the complex interactions of pathological, social and psychosocial factors (*WHO, 2020*). According to previous studies, people who suffered mental health problems such as low self-esteem, loneliness, unmet social support, entrapment, defeat and dissatisfied interpersonal needs were more vulnerable to depression. Self-esteem had a close relationship with the onset, maintenance, and relapse of depression (*Choi et al., 2019*; *Han & Kim, 2020*; *Sowislo & Orth, 2013*). The relationship between loneliness and depression was longitudinal effect which is bidirectional but more significant with former as origin (*Domènech-Abella et al., 2019*). Meanwhile, social support was also significantly correlated with depression and partially mediated loneliness and depression (*Liu, Gou & Zuo, 2016*). The roles of defeat and entrapment in depression were first proposed by *Gilbert & Allan (1998)* and had been confirmed to be promising variables for the study of depression (*Carvalho et al., 2013*; *Gilbert & Allan, 1998*; *Trachsel et al., 2010*). The interpersonal theory of suicide had been applied in prediction of future depression with thwarted belongingness and perceived burdensomeness shown to give contribution to higher interpersonal shame and eventually led to depression (*Carrera & Wei, 2017*). The roles of the psychosocial factors above in the development of depression are worth exploring in patients with STIs.

STIs patients may experience multiple mental health problems that may contribute to depression vulnerability in a syndemic way. The concept of syndemic was first proposed by *Singer & Clair (2003)* to describe that several concurring epidemics may synergistically interact to undermine health in a certain population (*Guadamuz et al., 2014*; *Tsai, 2018*). Synergistic epidemic or syndemic effect is the aggregation of more than two concurrent or sequential epidemic or disease clusters in a population. The syndemic effect exacerbates

the disease burden. The syndemic theory has become increasingly popular for policymaker and program implementers to improve population health, especially in the field of HIV treatment and prevention (*Tsai, 2018*). Previous studies found the combined impacts of psychosocial problems on suicide ideation in HIV+ patients (*Wang et al., 2018*), men who have sex with men (MSM) (*Li et al., 2016*) and STIs patients (*Wang et al., 2020*). Likewise, we hypothesized that there may be magnifying effects of the syndemic of multiple psychosocial factors on depression among STIs patients.

Data reflecting the prevalence of mental health problems, their association with depression and comprehensive effects on depression among STIs patients are limited. The purposes of our study were: (1) to gain in-depth understanding of the STIs patients' mental health status; (2) to construct a comprehensive framework about the combined effects of mental health issues on depression by applying syndemic theory; (3) to develop a preliminary psychosocial screening framework to effectively identify the prodrome of depression and to provide theoretical reference for psychosocial intervention strategies among STIs patients.

## MATERIALS & METHODS

### Participants and eligibility criteria

The cross-sectional study was launched in March, 2018. Patients from two branches (QiuJiang Rd. and BaoDe Rd., Jingan District) of Shanghai Skin Disease Hospital were recruited. Shanghai Skin Disease Hospital is one of the top hospitals specializing in management of STIs, with 136,146 outpatients and 1,845 discharged patients in 2020 ranking first in Shanghai.

Patients either walk-in or hospitalized were invited if they met the following criteria: (1) Aged 18 years or more; (2) clinically diagnosed as all forms of STIs. The STIs were classified into five common diseases (HIV/AIDS, syphilis, gonorrhea, genital warts, genital herpes) and others (*Chen, 2019*); (3) being able to read and sign informed consent and not participating in similar studies in the past six months. The exclusion criterion was participants with mental or cognitive impairment, unconsciousness so that they cannot verbalize their real feelings or fill out questionnaires.

Given that the prevalence of depression among STIs patients was 43% (*Mo et al., 2015*), using alpha of 0.05 and a relative error for sampling 0.05, taking into account a 30% non-response rate, required sample size was calculated to be 562. In our study, a total of 910 patients were finally recruited in the study.

### Ethics

The study has been approved by Public Health and Nursing Ethics Committee, Shanghai Jiao Tong University School of Medicine. Background information provided, written informed consent were signed by patients before collection of information. During the process of interview, participants were free to ask any question or to withdraw if they did not want to continue.

## Study procedure

The cooperation between our study team and Shanghai Skin Disease Hospital was based on a contract signed beforehand with all doctors who worked at STIs department, in-patients and out-patients recruited in the survey. Interviews were carried out by senior medical and graduate students from School of Medicine, Shanghai Jiao Tong University, who had received adequate trainings including in-person review, quality control strategies and privacy protection.

The doctors invited qualified patients to participate in the study. After receiving preliminary approval, the patients were invited to a separate room by the investigators. Our investigators informed participants the study goal, study procedure, potential risks and answered their questions and concerns about the study. Copy of questionnaire along with informed consent was provided for each eligible individual. Only after signing the informed consent, can participants completed self-filled questionnaires. A bonus of 80 RMB (US$12) was given to each person for his/her participation.

## Measures
## Sociodemographic variables

Sociodemographic variables included outpatient or inpatient, age group, gender, education level, marital status, residence status, monthly income, health insurance, self-reported sexual orientation, HIV status and type of STIs.

## Psychosocial variables
### Depression

The Patient Health Questionnaire-9 (PHQ-9) was adopted as a scale for depression evaluation in STIs patients. This 9-item self-report questionnaire was a widely used screening scale for depression in non-psychiatric settings, such as community, primary care and general hospitals (*Spitzer, Kroenke & Williams, 1999*). Severity of each symptom during the previous two weeks was marked from 0 (not at all) to 3 (nearly every day) (*Kroenke, Spitzer & Williams, 2001*). A total score (range: 0–27) of ≥10 was the most commonly- recommended cut-point for "clinically significant" symptoms (*Kroenke et al., 2010*; *Manea, Gilbody & McMillan, 2015*). Reliability and validity of its Chinese version had been confirmed by prior study (*Xing-Chen et al., 2014*). The Cronbach's $\alpha$ coefficient of PHQ-9 was 0.91 in this study.

### Self-esteem

Self-esteem was evaluated with Rosenberg Self-esteem Scale (RSES) (*Rosenberg, 1965*), a widely used tool containing 10 items which are scored on 4 points ranging from 0 (strongly disagree) to 3 (strongly agree). Higher scores indicated higher levels of global feelings of self-worth (range 0–30) (*Sinclair et al., 2010*). Any outcome lower than the norm of 15 was regarded as an indication of low self-esteem (*Li et al., 2016*; *Polat et al., 2015*). The scale was translated into Chinese by Cheung and Lau (*Cheung & Lau, 1985*) and was proved to have good reliability and validity (*Hu & Ai, 2016*). The Cronbach's $\alpha$ coefficient of RSES was 0.84 in this study.

### Social support

Social support from sources of family, friends and significant others was measured by Multidimensional Scale of Perceived Social Support (MSPSS) (*Zimet et al., 1988*), a 12-item scale. It was scored on a 7-point scale ranging from 1 (very strongly disagree) to 7 (very strongly agree), with a higher total score indicating higher social support (Cronbach's $\alpha = 0.95$, range 12–84). The reliability and validity of Chinese version of MSPSS had been verified (*Yuan Chen et al., 2018*). Any mark higher than the 25th percentile of 59 would indicate acceptable social support (*Li et al., 2016*).

### Loneliness

The USL-8 loneliness scale by Hays and DiMatteo was derived from the total 20 original questions at an attempt to shorten survey time and enhance answer quality (*Hays & Di Matteo, 1987*). Respondents scored their level of frequency with eight statements on a scale ranging from 1 (never) to 4 (always), with higher scores suggesting a higher degree of loneliness. (Cronbach's $\alpha = 0.81$, range 8–32). This study used a Chinese version of the ULS-8 Loneliness Scale. The translated version of the scale was created by Zhou and his colleagues (2012) for their own research on Chinese participants (*Zhou et al., 2012*). With no exiting recommendation for a cut-point for USL-8, we defined the 75th percentile of 18 as the cut-point of significant loneliness (*Li et al., 2016*).

### Entrapment

Entrapment Scale (ES) with 16 items quantified the escape motivation raised by outside world and internal feelings (Cronbach's $\alpha = 0.97$, range 0–64) (*Gilbert & Allan, 1998*). The response options for each item were "not at all," "a little bit," "moderately," "quite a bit," and "extremely,' which correspond to scores of 0–4 (*Griffiths et al., 2014*). The Chinese version of ES used in this study had good validity and reliability in medical students (*Ruijie Gong et al., 2019*). Any score higher than 75th percentile of 20 would be seen as high level of entrapment.

### Defeat

The level of defeat was presented by the 16-item Defeat Scale (DS) on ones' thoughts about themselves in the past week (Cronbach's $\alpha = 0.92$, range 0–64) (*Gilbert & Allan, 1998*). Individuals indicated the extent to their feeling with each of the items on a 5-point rating scale, ranging from 0 (never) to 4 (always) (*Griffiths et al., 2014*). The reliability and validity of the defeat scale in Chinese had been confirmed in medical students (*Hua Tang et al., 2019*). Participants who rated a sum higher than 75th percentile of 23 were classified into high sense of defeat.

### Interpersonal needs

The 15-item Interpersonal needs questionnaire (INQ-15), designed by professor Van Orden was used to analyze patients' perceived burdensomeness and thwarted belongingness in the past week (*Van Orden et al., 2008*). Respondents rated how often they felt a certain way ranging from 1 (not at all true for me) to 7 (very true for me). Higher numbers reflected higher levels of perceived burdensomeness and thwarted belongingness (Cronbach's

$\alpha = 0.85$, range 15–105). The Chinese version of INQ-15 was a valid and reliable measure instrument for assessing interpersonal needs in university students (*Xiaomin Li et al., 2015*). Likewise, score higher than 75th percentile of 49 was calculated as cut-point to define the dissatisfaction with interpersonal need.

## Statistical analysis

Statistical analysis was performed *via* SPSS version 25.0 for Windows. At first, descriptive statistics were adopted to get an overview of sociodemographic characteristics, prevalence of depression, and psychosocial variables. Then, univariate logistic regression was conducted to explore the association between sociodemographic characteristics and depression, between mental health problems and depression. The relationship between all six psychosocial variables and depression was evaluated though multivariable logistic regression after adjusting for all significant sociodemographic variables. After that, the correlations between psychosocial variables was described by Spearman correlation coefficient. Finally, to examine the psychosocial syndemic effect on depression, different groups were divided based on the number of mental health problems, so that a univariable logistic regression considering all significant sociodemographic variables can be carried out.

## RESULTS

Sociodemographic characteristics and their association with depression were presented in Table 1. An overall 910 interviewers completed the survey, the majority (67.5%) of which were from outpatient clinics. The average age was 38.72, with a standard deviation of 13.03. Ratio between male and female was 1:1.08. Of all the participants interviewed, 44.1% were local and 93.6% had a health insurance to pay the bill. More than half (61.8%) of them were married, followed by 30.4% unmarried, 6.3% divorced and 1.5% widowed. 86.8% reported heterosexuality. For education and monthly income level, 25.2% reported a highest education level was lower than high school and 16.3% earned less than an average of 3,000 yuan each month.

Six sociodemographic variables (age group, gender, education level, monthly income, marital status, residence status) were significantly associated with depression. Participants younger than 25 years old were about 4 times (OR = 4.12, 95% CI [2.03–8.37]) more susceptible to depression than those older than 60. Women had a higher risk of depression than men (OR = 1.65, 95% CI [1.17–2.34]). Compared to married group, unmarried status was a risk factor of depression (OR = 1.87, 95% CI [1.30–2.68]). The groups of above 6001 yuan per month was less likely to be depressed compared with the lowest group, and the lower the education level, the greater the likelihood of depression. Non-local people rose the possibility of depression approximately 2 times (OR = 1.75, 95% CI [1.23–2.50]) higher than local residents.

### Mental health problems and their correlation with depression among STIs patients

Patients' psychosocial health status was demonstrated in Table 2. Of the 910 participants, 17.9% reported having had depressive symptoms. Meanwhile, 9.6% was grouped as low

**Table 1** Sociodemographic characteristics and their association with depression among STIs patients in Shanghai, China.

| Sociodemographic characteristics | Number (%) | Had depression | |
|---|---|---|---|
| | | Number (%) | ORu (95% CI) |
| **Case** | | | |
| Inpatient | 296 (32.5) | 60 (36.8) | 1.26 (0.89–1.80) |
| Outpatient | 614 (67.5) | 103 (63.2) | 1 |
| **Age group (years)** | | | |
| <25 | 89 (9.8) | 39 (23.9) | 4.12 (2.03–8.37)* |
| 25–40 | 505 (55.5) | 70 (42.9) | 0.85 (0.46–1.59) |
| 41–59 | 228 (25.1) | 40 (24.5) | 1.13 (0.58–2.19) |
| ≥60 | 88 (9.7) | 14 (8.6) | 1 |
| **Gender** | | | |
| Female | 472 (51.9) | 101 (62.0) | 1.65 (1.17–2.34)* |
| Male | 438 (48.1) | 62 (38.0) | 1 |
| **Education level** | | | |
| Less than high school | 229 (25.2) | 52 (31.9) | 1.78 (1.19–2.65)* |
| High school | 202 (22.2) | 43 (26.4) | 1.64 (1.07–2.50)* |
| College degree or above | 479 (52.6) | 68 (41.7) | 1 |
| **Monthly income (RMB)** | | | |
| ≥120001 | 201 (22.1) | 16 (9.8) | 0.23 (0.12–0.42)* |
| 6001–12000 | 259 (28.5) | 41 (25.2) | 0.49 (0.30–0.80)* |
| 3001–6000 | 302 (33.2) | 65 (39.9) | 0.72 (0.46–1.13) |
| ≤3000 | 148 (16.3) | 41 (25.2) | 1 |
| **Marital status** | | | |
| Unmarried | 277 (30.4) | 2 (1.2) | 1.87 (1.30–2.68)* |
| Divorced | 57 (6.3) | 12 (7.4) | 1.56 (0.79–3.08) |
| Widowed | 14 (1.5) | 67 (41.1) | 0.98 (0.21–4.44) |
| Married | 562 (61.8) | 82 (50.2) | 1 |
| **Residence status** | | | |
| Non-local | 509 (55.9) | 109 (66.9) | 1.75 (1.23–2.50)* |
| Local | 401 (44.1) | 54 (33.1) | 1 |
| **Health insurance** | | | |
| Yes | 852 (93.6) | 150 (92.0) | 0.74 (0.39–1.41) |
| No | 58 (6.4) | 13 (8.0) | 1 |
| **HIV status** | | | |
| Positive | 94 (10.3) | 19 (11.7) | 1.23 (0.70–2.16) |
| Unknown | 370 (40.7) | 68 (41.7) | 1.10 (0.76–1.57) |
| Negative | 446 (49.0) | 76 (46.6) | 1 |
| **Self-reported sexual orientation** | | | |
| Non-heterosexual | 86 (9.5) | 15 (9.2) | 0.96 (0.53–1.72) |
| Not sure | 34 (3.7) | 5 (3.1) | 0.78 (0.30–2.05) |
| Heterosexual | 790 (86.8) | 143 (87.7) | 1 |

**Table 1** (*continued*)

| Sociodemographic characteristics | Number (%) | Had depression | |
|---|---|---|---|
| | | Number (%) | ORu (95% CI) |
| **Type of STIs** | | | |
| Gonorrhea | 30 (3.3) | 3 (1.8) | 0.56 (0.08–3.83) |
| Syphilis | 365 (40.1) | 68 (41.7) | 1.15 (0.25–5.35) |
| Genital warts | 361 (39.7) | 62 (38.0) | 1.04 (0.22–4.85) |
| Genital herpes | 28 (3.1) | 7 (4.3) | 1.67 (0.29–9.52) |
| Others | 114 (12.5) | 21 (12.9) | 1.13 (0.23–5.54) |
| HIV | 12 (1.3) | 2 (1.2) | 1 |

**Notes.**

ORu, Univariate odds ratios.

*$P < 0.05$.

**Table 2  Frequency distributions of psychosocial variables among STIs patients in Shanghai, China.**

| Psychosocial variables | Number (%) | Had depression Number (%) |
|---|---|---|
| **Depression** | | |
| High level (score $\geq$ 10) | 163 (17.9) | / |
| Low level (score < 10) | 747 (82.1) | / |
| **Self-esteem** | | |
| Low level (score < 15) | 87 (9.6) | 50 (30.7) |
| High level (score $\geq$ 15) | 823 (90.4) | 113 (69.3) |
| **Social support** | | |
| Low level (score < 59) | 237 (26.0) | 62 (38.0) |
| High level (score $\geq$ 59) | 673 (74.0) | 101 (62.0) |
| **Loneliness** | | |
| High level (score $\geq$ 18) | 263 (28.9) | 100 (61.3) |
| Low level (score < 18) | 647 (71.1) | 63 (38.7) |
| **Entrapment** | | |
| High level (score $\geq$ 20) | 236 (25.9) | 119 (73.0) |
| Low level (score < 20) | 674 (74.1) | 44 (27.0) |
| **Defeat** | | |
| High level (score $\geq$ 23) | 243 (26.7) | 113 (69.3) |
| Low level (score < 23) | 667 (73.3) | 50 (30.7) |
| **Interpersonal needs** | | |
| High level (score $\geq$ 49) | 232 (25.5) | 78 (47.9) |
| Low level (score < 49) | 678 (74.5) | 85 (52.1) |

level of self-esteem and 26.0% perceived inadequate social support. When using the 75th percentile of the scale scores as the cut points, 28.9% felt significantly lonely, 25.9% and 26.7% reported entrapment and defeat, respectively, and 25.5% had dissatisfied interpersonal needs.

Table 3 depicted the correlation between depression and other mental health problems. As seen in the Table 3, all our selected mental health problems showed significant
**Table 3** Descriptive analysis of psychosocial variables and their correlations with depression among STIs patients in Shanghai, China.

| Variables | Mean (SD) | Median (IQR) | 1 | 2 | 3 | 4 | 5 | 6 | 7 |
|---|---|---|---|---|---|---|---|---|---|
| 1.Depression | 6.19 (5.39) | 5 (7) | 1 | | | | | | |
| 2. Self-esteem | 19.8 (4.64) | 19 (6) | −0.38* | 1 | | | | | |
| 3. Loneliness | 14.72 (4.88) | 14 (8) | 0.48* | −0.50* | 1 | | | | |
| 4. Social support | 65.48 (14.67) | 69 (17) | −0.16* | 0.33* | −0.41* | 1 | | | |
| 5. Entrapment | 12.12 (13.30) | 7.5 (20) | 0.71* | −0.38* | 0.53* | −0.23* | 1 | | |
| 6. Defeat | 16.50 (11.15) | 14 (15) | 0.60* | −0.61* | 0.60* | −0.30* | 0.38* | 1 | |
| 7. Interpersonal needs | 36.33 (14.63) | 35 (25) | 0.34* | −0.54* | 0.57* | −0.42* | 0.42* | 0.58* | 1 |

Notes.

*$p < 0.01$.

relationships with depression, among which entrapment presented the strongest association with depression ($r = 0.71$, $P < 0.01$). Loneliness ($r = 0.48$, $P < 0.01$), entrapment, defeat ($r = 0.60$, $P < 0.01$) and interpersonal needs ($r = 0.34$, $P < 0.01$) were positively correlated with depression, whereas self-esteem ($r = −0.38$, $P < 0.01$) and social support ($r = −0.16$, $P < 0.01$) were negatively related with depression.

## Mental health problems associated with depression

The results of binary regression were shown in Table 4. After adjusting for age group, gender, education level, monthly income, marital status, residence status, all psychosocial variables included were statistically significant with depression. The low levels of self-esteem (adjusted odds ratio [AOR]: 6.74, 95% CI [4.08–11.14]) and social support (AOR:1.73, 95% CI [1.18–2.54]), the high levels of loneliness (AOR: 5.75, 95% CI [3.96–8.36]), entrapment (AOR: 14.80, 95% CI [9.69–22.59]), defeat (AOR: 12.04, 95% CI [7.87–18.44]), interpersonal needs (AOR: 3.35, 95% CI [2.28–4.92]) were at elevated risk for depression. However, the multivariate logistic regression only remained four significant variables: self-esteem (Odds ratios obtained from forward stepwise multivariate logistic regression using significant variables of the univariate analysis as input [ORm]: 2.18, 95% CI [1.19–4.00]), social support (ORm: 2.18, 95% CI [1.37–3.46]), entrapment (ORm: 6.31, 95% CI [3.75–10.62]), defeat (ORm: 2.60, 95% CI [1.51–4.48]).

## Verification of the syndemic effect of mental health problems

The results of the association between the number of syndemic condition and depression were presented in Table 5. It was found that having two or more psychosocial problems at the same time (37%, 337/910) magnified effect in fusing depression (AOR:11.94, 95% CI [7.70–18.53]). The participants were further divided into non-syndemic group, low-level syndemic group (have two to three psychosocial problems) and high-level syndemic group (have four or more psychosocial problems). The low-level group (AOR: 7.52, 95% CI [4.60–12.28]) and high-level group (AOR: 22.12, 95% CI [13.19–37.09]) showed a prominent syndemic effect compared with those in the non-syndemic group.

**Table 4  Mental health problems associated with depression among STIs patients in Shanghai, China.**

| Mental health problems | ORu (95% CI) | AOR (95% CI) | ORm (95% CI) |
|---|---|---|---|
| **Self-esteem** | | | |
| High level | 1 | 1 | 1 |
| Low level | 8.42 (5.31–13.57)** | 6.74 (4.08–11.14)** | 2.18 (1.19–4.00)* |
| **Social support** | | | |
| High level | 1 | 1 | 1 |
| Low level | 2.01 (1.40–2.87)** | 1.73 (1.18–2.54)** | 2.18 (1.37–3.46)** |
| **Loneliness** | | | |
| Low level | 1 | 1 | |
| High level | 5.69 (3.97–8.15)** | 5.75 (3.96–8.36)** | |
| **Entrapment** | | | |
| Low level | 1 | 1 | 1 |
| High level | 14.56 (9.78–21.69)** | 14.80 (9.69–22.59)** | 6.31 (3.75–10.62)** |
| **Defeat** | | | |
| Low level | 1 | 1 | 1 |
| High level | 10.73 (7.31–15.73)** | 12.04 (7.87–18.44)** | 2.60 (1.51–4.48)** |
| **Interpersonal needs** | | | |
| Low level | 1 | 1 | |
| High level | 3.53 (2.48–5.04)** | 3.35 (2.28–4.92)** | |

**Notes.**

ORu, Univariate odds ratios; AOR, Odds ratios adjusted for age group, gender, education level, monthly income , marital status, residence status.; ORm, Odds ratios obtained from forward stepwise multivariate logistic regression using significant variables of the univariate analysis as input.

*$p < 0.05$.

**$p < 0.01$

**Table 5  Association between the number of syndemic condition and depression among STIs patients in Shanghai, China.**

| | Number (%) | Had depression | |
|---|---|---|---|
| | | Number (%) | AOR (95% CI) |
| **Have a syndemic** | | | |
| No (have no more than one psychosocial problem) | 573 (63.0) | 31 (19.0) | 1 |
| Yes (have two or more psychosocial problems) | 337 (37.0) | 132 (81.0) | 11.94 (7.70–18.53)** |
| **Number of syndemic conditions** | | | |
| No (have no more than one psychosocial problem) | 573 (63.0) | 31 (19.0) | 1 |
| Low level (have two to three psychosocial problems) | 201 (22.1) | 59 (36.2) | 7.52 (4.60–12.28)** |
| High level (have four or more psychosocial problems) | 136 (14.9) | 73 (44.8) | 22.12(13.19–37.09)** |

**Notes.**

AOR, Odds ratios adjusted for age group, gender, education level, monthly income, marital status, residence status.

**$p < 0.01$.

# DISCUSSION

The prevalence of depression of STIs patients in our study was 17.9%. Compared with the prevalence of depression with 5.1% and 3.6% in ordinary women and men, respectively, the depression status of STIs patients deserved more attention. The concurrence of two or more mental health problems with a high prevalence of 37% demonstrated the syndemic

effect to magnify the depressive symptoms of the population. The more psychosocial problems STIs patients had, the more likely they were to have depression. The association between mental health issues may play important role in sydemic effect on depression.

We have found 6 possible sociodemographic factors contributing to depression among STIs patients. The depression rate of female patients was much higher than that of male patients (21.4% *versus* 14.1%, $P = 0.004$), which was consistent with an American study (51.9% *versus* 31.9%, $p = 0.023$) (*Erbelding et al., 2001*). Also, women had higher prevalence of STIs and depression than men in Canada (*Chen et al., 2008*). Unemployment and worsen relationship with sex partner after STIs diagnosis in female patients were risk factors of probable depression (*Mo et al., 2015*). How age got to become negative factor may be explained by higher sexual impulsivity, less awareness of the disease, lower ability to ask for help and higher vulnerability to events in the younger age group. Emotional and financial support from families and friends may explain married status and local residence as protective factors in depression development. Education degree and its related income levels, affecting medical resource, professional consultant and cost of medication can also be blocks to psychosocial health.

In traditional belief, sexually transmitted infections were considered as social stigma and a shame not only to individual but also to the family (*Lichtenstein, Hook 3rd & Sharma, 2005*). The consequent stress as well as physical discomfort may provoke poor mental well-being (*Duncan et al., 2001*). In our study, all 6 psychosocial variables were related to depression, especially self-esteem, social support, entrapment and defeat played significant roles in increasing the risk of depression. Self-esteem and social support had been regarded as important resources to fight against stress and to maintain health (*Thoits, 2010*). According to the vulnerability model of depression (*Orth & Robins, 2013*), lower self-esteem can put individuals at a higher risk of depression when confronting significant stressors. Consistent with the vulnerability model, it had been shown in a meta-analysis of 95 longitudinal studies that low self-esteem carried significantly stronger effect on depression ($\beta = -0.16$) (*Sowislo & Orth, 2013*). Studies had also verified the perception of social support (available assistance, receiving actual help and integration in a social work) can prevent the risk of depression (*Liu, Gou & Zuo, 2016*; *Maeda et al., 2013*; *Sonnenberg et al., 2013*). Usage of Internet was encouraged by a study on social media who discovered that the more social support people thought they perceived from social media, the less likely they would be trapped in depression (*Park et al., 2016*). The social support intervention by social media can be carried out in STIs patients to ease the mental health burden.

The results found, in agreement with previous studies, that entrapment and defeat were important predictors of depression, which was also applicable to STIs patients. The concept of entrapment (an inability to escape from adverse situations) and defeat (failed social struggle) derived from the social rank theory of depression (*Gilbert & Allan, 1998*). Many of evolutionary theories noted that "failed struggle" can be considered as the essence of depression onset. The feelings of entrapment and defeat can emerge from unacceptable or relatively novel issues, social threat, stressors. To our knowledge, these are common problems encountered by STIs patients. The previous studies had shown clear and robust correlation between entrapment, defeat and depression (*Gilbert & Allan,*

*1998*; *Griffiths et al., 2014*; *Taylor et al., 2011*; *Trachsel et al., 2010*). Furthermore, a previous systematic review reported that self-perceived defeat and entrapment played key roles in depression, anxiety, suicidal tendency, and post-traumatic stress disorder, and emphasized that entrapment played a decisive role in depression (*Siddaway et al., 2015*). There were implications for targeting perceptions of entrapment and defeat within psychosocial interventions for STIs patients suffering from depression and screening the population to identify those at risk of developing depression.

The most significant finding of the study was that psychosocial syndemic revealed a magnifying effect in predicting depression status of STIs patients. The interaction between psychosocial variables led to the coexistence of psychosocial problems. The self-esteem antecedent model argued that self-esteem had an effect on social support. Specifically, higher self-esteem can get individuals to develop positive social support networks because of their belief in their social worth, while those with lower self-esteem had difficulties in establishing positive social support systems because they usually avoided social interaction due to their worry about being rejected by others (*Marshall et al., 2014*). On the other hand, the self-esteem consequence model pointed out that positive social support can produce higher self-esteem (*Leary, 2005*). The mediation effect of social support on loneliness and depression can shed light on the concurrence effects of loneliness and social support on depression (*Liu, Gou & Zuo, 2016*). In addition, some researcher conducted factor analysis to verify entrapment and defeat were best defined as on factor, suggesting that the perceptions cooccurred and led to development of mental disorders (*Griffiths et al., 2015*; *Griffiths et al., 2014*). Therefore, we integrated various mental health issues and formed a comprehensive framework for the preliminary evaluation of depression among STIS patients. As a conclusion, a high number of psychosocial problems indicated a greater potentiality of depression.

### Limitations

This survey had several limitations. First, analysis of cross-sectional result alone was hard to figure out the causality thus a prospective study assessing the cause-and-effect relationship between psychosocial variables and depression is needed in the future. Second, the data in this study was only from Shanghai Skin Disease Hospital and obtained by convenient sampling method, so multicenter research nationwide with random sampling method is expected to further generalize the conclusion. Third, despite regular training and active feedback with interviewers, the bias of recall and self-report from interviewees were unavoidable. Fourth, some psychosocial problems related to STIs patients such as stigma and stress were not taken into consideration. Finally, there was insufficient evidence to offer generally recognized cut-offs to divide high-level and low-level groups of psychosocial variables.

## CONCLUSIONS

Our study reviewed the contribution of each psychosocial variable to the development of depression in sexually transmitted infections patients and extended the literature into the study. The correlation between psychosocial problems and depression was confirmed, and

a syndemic effect of psychosocial problems on increasing odds of depression was identified. We emphasize the importance of early and integrated psychosocial intervention to alleviate their mental burden so that the occurrence and progression of depression can be blocked.

## ACKNOWLEDGEMENTS

We are grateful to the individuals who volunteered their time to participate the study.

### Funding
This work was supported by the National Key R&D Program of China (Grant Nos. 2020YFC2006400), Shanghai Three-year Action Plan for Public Health under Grant GWV-10.2-XD13, GWV-10.1-XK15, GWV-10.1-XK18, and by Strategic collaborative innovation team (SSMU-ZLCX20180601). The funders had no role in study design, data collection and analysis, decision to publish, or preparation of the manuscript.

### Grant Disclosures
The following grant information was disclosed by the authors:
National Key R&D Program of China: 2020YFC2006400.
Shanghai Three-year Action Plan for Public Health: GWV-10.2-XD13, GWV-10.1-XK15, GWV-10.1-XK18.
Strategic collaborative Innovation Team: SSMU-ZLCX20180601.

### Competing Interests
The authors declare there are no competing interests.

### Author Contributions
- Chen Xu and Yuan Shi performed the experiments, analyzed the data, authored or reviewed drafts of the paper, and approved the final draft.
- Xiaoyue Yu performed the experiments, analyzed the data, prepared figures and/or tables, and approved the final draft.
- Ruijie Chang, Huwen Wang, Hui Chen, Rongxi Wang, Yujie Liu and Shangbin Liu performed the experiments, prepared figures and/or tables, and approved the final draft.
- Yong Cai, Yang Ni and Suping Wang conceived and designed the experiments, authored or reviewed drafts of the paper, and approved the final draft.

### Human Ethics
The following information was supplied relating to ethical approvals (i.e., approving body and any reference numbers):
Public Health and Nursing Ethics Committee, Shanghai Jiao Tong University School of Medicine approved this study.

### Data Availability
Raw measurements are available as a Supplementary File.

## Supplemental Information

Supplemental information for this article can be found online at http://dx.doi.org/10.7717/peerj.12022#supplemental-information.

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
