# Peer review of "The syndemic condition of psychosocial problems related to depression among sexually transmitted infections patients"

_PeerJ, doi:10.7717/peerj.12022_

## Round 0.1 · original submission · Major Revisions

The reviewers have provided a number of recommendations that I ask you to seriously consider, and respond to each one.

·

Basic reporting

no comment

Experimental design

I think the author might consider two more factors: the type of STI and the severity of the STI.
Since they recruited participants with different STIs...

Does the severity of STI have an impact on depression tendency?
Is it possible that certain type of STI is particularly associated with depression? If so, what is the possible explanation?

Validity of the findings

no comment

Additional comments

no comment

Reviewer 2 ·

Basic reporting

Hello authors, I hope you're all safe and well. Overall the article is well written. Only few minor changes are required.

The references, mentioned at the end of their respective paragraphs/sentence, are written as for example (Hans and Kim, 2020). According to APA format 'and' is only written when the reference is mentioned in the sentence or para for example Hans and Kim (2020) mentioned....

Corrections that need to be made for those references have been provided (Details for some have been given form more clarity):

Line 71: (Han & Kim, 2020)
Line 72: (Sowislo & Orth, 2013)

Line 76: The reference in this line has been written as Gilbert and Allan in 1998 (Gilbert and Allan, 1998). The correct format has been provided which is:

first proposed by Gilbert and Allan (1998) and has been confirmed

The sentence from the article has been mentioned as well to provide more clarity of the correction.

Line 80: (Carrera & Wei, 2017)
Line 166: Cheung and Lau (1985)
Line 167: (Hu & Ai, 2016)
Line 180: (Hays & DiMatteo, 1987)

Line 184: In this line either the authors can mention the reference as Zhou and his colleagues (2012) or mention it as Zhou et al. (2012). Or rephrase it in a manner that they provide the reference at the end of the respective para i.e (Zhou et al., 2012). However according to the APA format they cannot mention both formats.

For example
This study uses a Chinese version of the ULS-8 Loneliness Scale. The translated version of the scale was created by Zhou and his colleagues (2012) for their own research on Chinese participants.

Line 190: (Gilbert & Allan, 1998)
Line 198: (Gilbert & Allan, 1998)
Line 205: Van Orden et al. (2008)
Line 308: play (it was misspelled as paly)
Line 310-311: (Orth & Robins, 2013)
Line 314: (Sowisho & Orth, 2013)
Line 324: (Gilbert & Allan, 1998)
Line 329: (Gilbert & Allan, 1998)
Line 335: 'developing depression' instead of 'development depression'
Line 345: ‘can’ instead of ‘and’
Line 348-349: (Griffiths, 2014; Griffiths, 2015)

Line 386: References

All the references mentioned under this line need to be corrected into APA format unless another format is being used, as required by the journal in which the article will be published. If the latter is the case then all the references within the article needs to be consistent with that format. Incase APA pattern is being used then refer to the APA manual 6th edition to make the necessary corrections

In the following feedback is provided for background and context:

Line 111-112: Although researches of the most common STIs have been provided, the study included participants with all forms of STIs. Mention the rationale for this using other researches.

Also in Table 1 titled; Sociodemographic characteristics and their associations with depression among STIs patients in Shanghai, China

Under the heading 'Types of STI'

Various STIs have been mentioned in the category of others

Provide the process to explain how the STIs were chosen to come under this category with reference to previous researches.

Line 149: Provide a rationale for the use of DSM-IV criteria based test when DSM-5 exists

Since DSM-5 is now in use for the purpose of diagnoses and tests like Beck Depression Inventory (BDI-II), a screener, has also been updated in context of the symptoms provided by the DSM-5. A concrete rationale needs to be provided as to why this scale was used with reference to research.

Line 236-237: Six sociodemographic variables (age, gender, education, income, marital status, residence status) have significant effects on depression.

Provide research references for each of the demographics, mentioned.

Line 338: The coexistence of psychosocial problems is evidenced based on previous studies.

Mention references for those previous studies.

Line 338-339: The self-esteem antecedent model argues that self-esteem has an effect on social support

Provide reference for self-esteem antecedent model

Line 343: Provide reference for self-esteem consequence model

Experimental design

no comment

Validity of the findings

no comment

---

## Round 0.2 · Minor Revisions

Thank you for your modifications. I would now like to ask that your odds ratios, 95% confidence limits, and correlation coefficients have no more than two decimal places. Three decimal places is excessive given your sample sizes.

---

## Round 0.3 · accepted · Accept

Thank you for making these final requested changes.